# Urban Esthetic Benefits of Undergrounding Utility Lines in Consideration of the Three-Dimensional Landscape

**Shota Ishigooka [1], Tatsuhito Kono [2,*] and Hajime Seya [3,†]**

1. Graduate School of Engineering, Tohoku University, Aoba 468, Aoba-ku, Sendai 980-0879, Japan; ishigoka@plan.civil.tohoku.ac.jp
2. Graduate School of Information Sciences, Tohoku University, Aoba 06, Aoba-ku, Sendai 980-0879, Japan
3. Department of Civil Engineering, Graduate School of Engineering, Kobe University, 1-1 Rokkodai-cho, Nada-ku, Kobe 657-8501, Japan; hseya@people.kobe-u.ac.jp
* Correspondence: kono@plan.civil.tohoku.ac.jp
† Alphabetical listing of authorship.

**Abstract:** Since the relative weight of form to function has been increasing in urban planning, from the esthetic viewpoint, many cities in the world have been actively pursuing the undergrounding of overhead utility lines. Esthetic factors are urban externalities in the sense that they are not directly traded in markets. Therefore, we need to control them optimally based on their benefits. In this study, we appraise the benefits of undergrounding utility lines in Japan and clarify the dependency of the residents' willingness to pay (WTP) on the road width and building height. Our results show that the WTP for undergrounding utility lines is lower as the road becomes wider and the buildings along the road become higher. However, when the road is wide, the WTP does not change much regardless of the height of the buildings. In addition, the average value of the benefit–cost ratios of previous undergrounding projects is from approximately 2.27 to 2.65. However, 3–17% of these projects have benefit–cost ratios of less than 1.

**Keywords:** urban esthetic project; undergrounding utility lines; hedonic approach; road width; building height

**JEL Classification:** H41; H43; R38

## 1. Introduction

In recent urban planning and development, the relative weight of form to function has been increasing in many countries. One issue with urban esthetics in many areas all over the world is overhead electricity distribution and telecommunication lines. In addition, the associated poles are often obstacles to transportation. Such esthetic and public aspects of overhead electricity lines and poles are externalities in terms of economics because they are not directly traded in markets.

Although many countries have been actively pursuing the undergrounding of overhead utility lines, undergrounding rates are still low. For example, even in European countries as a whole, the undergrounding ratios are low (e.g., 39% in France and 33% in Italy), although they prefer beautiful scenery, and, indeed, many large European cities have almost completely removed the lines and poles (e.g., about 100% in London, Paris, and Hamburg).

Outside Europe, there are many cities with low undergrounding rates (e.g., 65% in Washington DC, 49% in Anaheim in the US, 17% in Ho Chi Minh in Vietnam, and 49% in Seoul in Korea). Among them, particularly in Japan, even large cities still have very low undergrounding rates. According to the Ministry of Land, Infrastructure, Transport and Tourism (MLIT) [1], the rate in the 23 special wards of Tokyo is 8%, and in Osaka, it is 6%.

The cost of undergrounding lines in Japan is high. Indeed, according to a survey of the Japan Association of City Mayors by the MLIT [2] on undergrounding utility lines, 186 out of the 196 local governments answered that the cost is the main obstacle to progress in undergrounding utility lines. Because of such high costs, it may be unrealistic to underground all of the utility lines. Instead, we need to discuss whether or not each individual line is worth being undergrounded.

As for previous studies in Japan, Adachi and Inoue [3] employed the hedonic approach with the land price data in Osaka and found that undergrounding utility lines increases land prices by 17,000 JPY/m$^2$. Using the difference-in-differences approach, Oba [4] finds that, in Kyoto, land prices increase by 12.5% within 50 m of the undergrounding site (which corresponds to 92,000 JPY/m$^2$) and by 7.5% within 200 m.

Applying the hedonic approach to the data in Canberra, McNair and Abelson [5] found that undergrounded lines increase house prices by 2.9%. Using their land price data and the average lot area in Canberra of 220 m$^2$, the increased land price can be calculated as 5600 JPY/m$^2$. Using contingent valuation methods, McNair et al. [6] found that a household would be willing to pay about 1.2 million yen (=A\$16,000) or more for undergrounding in suburbs. Hamilton and Schwann [7] concluded that, applying the cross-sectional hedonic approach to the property value data in metropolitan Vancouver, the properties adjacent to an overhead line lost 6.3% of their values. Sims and Dent [8] estimate that the values of the property within 100m of an overhead line is reduced by 6–17% (an average of 11.5 percent), using UK data. François [9] shows that using the property value data of the City of Brossard, Canada, between 1991 and 1996, an overhead line leads to a significant drop in property value of 9.6% on average due to the visual encumbrance. Navrud et al. [10] estimated the willingness to pay (WTP, hereafter) for undergrounding cables using the contingent valuation method. The WTP per household is estimated at about 732 to 1988 Norwegian krone (=0.12 euro) depending on the distance from the house to the power line. Besides, Kaliampakos et al. [11], Qiao et al. [12], Mavrikos and Kaliampakos [13], and Peng et al. [14] reviewed and estimated the benefits of underground space utilization from the viewpoint of landscape improvements, although these studies do not focus on undergrounding utility lines specifically.

Following most of the previous approaches, we adopt the hedonic approach with cross-sectional data. Another possible approach includes a difference-in-differences method (e.g., Deschens and Greenstone [15]). However, this would require multiple time period data and thus it is difficult to apply this approach without further data collection. The current paper is the first to evaluate the benefit, using the nationwide data of undergrounded lines all over Japan. More importantly, we focus on the dependency of the values of undergrounding projects on the current landscape as follows. In Japan, as well as many other countries, the property tax rate is fixed regardless of whether utility lines are undergrounded or not. Therefore, the undergrounding costs are not capitalized in land prices.

We captured the effects of the three-dimensional urban landscape on the housing prices and the residents' WTP in residential areas, using Japanese data. Here, the three-dimensional urban landscape consists of the width of a road and the height of buildings. In terms of the benefits of landscape, residents walk on the streets in their neighborhood so they might enjoy the roads with undergrounded lines. The increase in the expanse of visible sky with undergrounded utility lines depends on road width and building height. In addition, traffic obstruction by utility lines depends on the width of a road. No previous papers have considered the effects of road width and building height on the WTP for undergrounding utility lines.

We identified 1591 undergrounded lines in residential areas before 2016 in 33 Japanese prefectures. Our analysis is composed of two analyses. First, we explore the effects of road width and building height on WTP for undergrounding utility lines. Second, we appraise the benefit–cost ratios (B/C) of undergrounded utility lines in Japan.

The remaining sections are organized as follows. Section 2 theoretically explores the relationship between land prices, benefits, and the WTP of a household. Section 3

describes the empirical model and the dataset, and Section 4 explains our estimation results. Section 5 concludes this study.

## 2. Evaluating Benefits of Undergrounding Lines with the Hedonic Approach

*2.1. The Relation between Housing Prices and the WTP*

We estimate the change in housing prices due to the undergrounding of utility lines and the WTP for it. This section reviews the relationship between land prices, housing prices, and the WTP for a land improvement project, using the methods of Pines and Weiss [16], and discusses it from the viewpoint of undergrounding lines. Although many papers including those from Scotchmer [17] and Roback [18] analyze this relationship, the basic structure used here is identical to that of Pines and Weiss.

We extend Pines and Weiss' [16] function of a single amenity factor to include various amenities denoted by a vector $\mathbf{Z} = (z_1, z_2, \ldots, z_K)$. Because the utility level $U(x, q, \mathbf{Z})$, where $x$ is the consumption of composite goods and $q$ is lot size consumption, is common among locations under a free migration assumption, Equation (1) holds:

$$\frac{dU(x, q, \mathbf{Z})}{d\mathbf{Z}} \cdot d\mathbf{Z} = 0 \tag{1}$$

where $\cdot$ expresses the inner product, and the variables before and after $\cdot$ are in a vector.

Equation (1) implies that if people migrate to another location with a different level of the amenity factor $\mathbf{Z}$, then the same utility level is attained. Total differentiation of Equation (1), with optimal condition for households $\partial U/\partial q / \partial U/\partial x = r$, which indicates the equality of the marginal substitution and the lot rent, and the total differentiating the budget constraint, $w = x + rq$, with respect to Z yields

$$\frac{\partial U/\partial \mathbf{Z}}{\partial U/\partial x} \cdot d\mathbf{Z} = q\frac{\partial r}{\partial \mathbf{Z}} \cdot d\mathbf{Z} \tag{2}$$

where $w$ is wage, and $r$ is lot rent. In Pines and Weiss, the target houses are only detached houses, not condominiums. Accordingly, lot rent, $r$, is identical to land rent. To derive Equation (2), $\frac{\partial w}{\partial \mathbf{Z}} = 0$ is assumed, whereas the original Pines and Weiss paper does not assume this. We only target residential areas, so this assumption at least approximately holds.

Equation (2) can be used for measuring the WTP for amenities. Actually, their main target is not to obtain this equation. Their target is to obtain the social benefit function, which is expressed by Equation (20) of the Pines and Weiss paper. $\partial U/\partial \mathbf{z} / \partial U/\partial x$. $\partial U/\partial \mathbf{z} / \partial U/\partial x$ implies the personal benefit of a marginal increase in amenities in terms of the composite goods. This shows that the WTP for amenities can be calculated as the multiplication of $\partial P/\partial z$ and $q$. We use this equation in Section 4.2.

As described above, we assume a homogeneous utility function. In reality, people who like beautiful scenery would like to live in an area with utility lines undergrounded. To account for this situation, we must consider the spatial sorting of heterogenous people over separate areas. However, as Section 3 shows, Japan has not had sufficient undergrounding sites to analyze the heterogeneity in residents. Accordingly, we could not account for this sorting for the time being. As Kanemoto [19] and Rosen [20] show, when people are heterogeneous, the market land price function is the envelope of the bid land price functions of various people. In this case, our estimation indicates the maximum values of undergrounded utility lines.

The WTP captured by the hedonic approach does not necessarily reflect all the benefits of undergrounding even in residential areas. For example, if visitors enjoy walking on the roads with undergrounded lines in a residential area, this benefit is not reflected in land prices. Measuring these uncaptured benefits remains a possibility for future research.

*2.2. Taking Account of Condominiums*

Equation (2) can measure the WTP per household living in a detached house. However, land can be occupied by condominiums. The WTP of a household is reflected in floor

space rent directly but it is not reflected in land rent in a simple way because a number of households share the land in the case of condominiums. Surprisingly, in the context of the hedonic approach, condominiums have not been explicitly taken into account.

We explore the relationship between floor rent and land rent. First, we set developers' behaviors as assumed by Joshi and Kono [21], Brueckner et al. [22], and Kono and Joshi [23]. Developers construct condominiums using capital and land. The cost function of supplying the floor area is given as $F(S)$, where $S$ is capital/land ratio, which implies building height. Setting the inverse function of $F(S)$ as $S(F)$, $S(F)$ implies the capital necessary for constructing the floor area $F$. The profit of a perfectly-competitive developer in zone $i$ is defined as

$$\Pi^i = F^i R^i - S(F^i) - P^i, \tag{3}$$

where $\Pi$ is the profit, $R$ is floor rent, and $P$ is the land rent. The price of capital is normalized at one.

The profit of the developer is maximized but is zero because of the perfect competition among developers, that is, $\Pi^i = 0$. Landowners rent their land to the highest bidders. Thus, developers maximize the land rent as follows.

$$\max_{F^i} \quad P^i = F^i r^i - S(F^i) \tag{4}$$

Since the first order condition is $r(i) - \partial S / \partial F = 0$, $F^i$ and $S(F^i)$ are functions of $r^i$. Accordingly, substituting these into Equation (4), we have

$$P^i = F(r^i) r^i - S(r^i) \tag{5}$$

To see the change in land prices with respect to amenities, we differentiate Equation (5) with respect to amenities $Z$.

$$\frac{dP^i}{dZ} = F(r^i) \frac{\partial r^i}{\partial Z} \tag{6}$$

Equation (6) is obtained by assuming that $F^i$ is variable. However, floor area ratio regulation is common in urban areas. In this situation, $F^i$ is fixed at a regulated level if the regulation is binding. Regarding optimal setting of floor area ratios, see Pines and Kono [24], Kono and Kawaguchi [25], Kono and Joshi [26], Yoshida and Kono [27], and Domon et al. [28]. Even in this situation, the same equation can be obtained directly from Equation (5). Indeed, Equation (4) in Brueckner et al. [22], which can be expressed as $dP^i/dA = \overline{F} \partial r^i / \partial A$ in our notation, expresses this binding case.

Multiplying both sides by $q^i$, dividing them by $F$, and exchanging the right-hand side with the left-hand side, yields

$$q^i \frac{\partial r^i}{\partial Z} = \frac{q_i}{F(r^i)} \frac{\partial P^i}{\partial Z} = \frac{1}{n^i} \frac{\partial P^i}{\partial Z} \tag{7}$$

where $n^i$ is household density, which is equal to $F(r^i)/q_i$. Substituting the left-hand side of Equation (7) into the right-hand side of Equation (2), we obtain the WTP measurement function in the case of condominiums as

$$\frac{\partial U/\partial \mathbf{z}}{\partial U/\partial x} \cdot d\mathbf{Z} = \frac{1}{n^i} \frac{\partial P^i}{\partial \mathbf{Z}} \cdot d\mathbf{Z} \tag{8}$$

Equation (8) is intuitively interpreted as follows. In the case of condominiums, multiple households residing in a condominium share the land. Accordingly, an increase in the land price is related to the multiple households' WTPs for an increase in local amenities. Therefore, household density $1/n^i$ is used in the case of condominiums, instead of lot size $q$ in the case of detached houses. However, this formula can be applied to detached houses as it is because $1/n^i = q$ where the building is occupied by only one household.

## 3. Empirical Model and Data

### 3.1. Empirical Model

The functional form of hedonic models cannot be pre-determined. We adopted the following three typical functional forms, that is, linear (Equation (9)), full log (Equation (10)), and semi Box-Cox (Equation (11)). These three forms have a common function $\Phi$, which expresses the effects of undergrounding utility lines. Note that dummy variables in Equation (10) take exp(1) or exp(0), whereas dummy variables in Equations (9) and (11) take 1 or 0.

$$P_i = \alpha_0 + \sum_{k=1}^{K} \alpha_k z_{ik} + \Phi + \varepsilon_i \quad (i = 1, 2, \cdots, n) \tag{9}$$

$$\ln(P_i) = \alpha_0 + \sum_{k=1}^{K} \alpha_k \ln(z_{ik}) + \Phi + \varepsilon_i \quad (i = 1, 2, \cdots, n) \tag{10}$$

$$\frac{P_i^\lambda - 1}{\lambda} = \alpha_0 + \sum_{k=1}^{K} \alpha_k z_{ik} + \Phi + \varepsilon_i \quad (i = 1, 2, \cdots, n) \tag{11}$$

$$\Phi = \delta_{iu} \left( \beta_u + \sum_{w=1}^{K} \beta_w \delta_{iw} + \beta_h \delta_{ih} + \sum_{m=1}^{K} \beta_m \delta_{im} + \sum_{p=1}^{K} \beta_p \delta_{ip} \right) + \beta_a \delta_{ia} \tag{12}$$
$$(i = 1, 2, \cdots, n)$$

where $P_i$ is land price per m$^2$ at location $i$, $\alpha_0$ is the intercept, $Z_{ik}$ is the $k$th ($k = 1, 2, \ldots, K$) attribute for location $i$, $\varepsilon_i$ is an error term, $\alpha_k$ is the regression coefficient for the $k$th attribute, $\delta_{iu}$ is the dummy variable representing whether the utility lines on the road are undergrounded or not. $\delta_{iw}$ is the dummy variable for road width. $\delta_{ih}$ is the dummy variable for building height. $\delta_{im}$ is the dummy variable of the multiplier effects of road width and building height. $\delta_{ip}$ is a prefecture dummy variable. $\delta_{ia}$ is the dummy variable representing the roads in the neighborhood not facing roads with undergrounded lines. $\beta_u$, $\beta_w$, $\beta_h$, $\beta_m$, $\beta_p$, and $\beta_a$ are regression coefficients for the dummy variables.

$\Phi$, which expresses the effects of undergrounding utility lines, is the most important function for our purposes. The dummy variable $\beta_u$, called "basis" in our paper, expresses the effect of the basic combination of road width and building height. Without loss of generality, the basis is set as the combination of narrow road and low buildings.

The dummy variable for road width $\delta_{iw}$ represents whether road width is narrow, medium, or wide (narrow, less than 5.5 m; medium, more than 5.5 m and less than 13.0 m; wide, more than 13.0 m). In terms of the dummy variable for building height, the current study uses the regulation value of the floor area ratio (FAR). The FAR is defined as the ratio of total floor space to lot area. Therefore, as the FAR becomes greater, higher buildings can be built. The floor area ratio is determined at 0.5, 0.6, 0.8, 1.0, 1.5, 2.0, 3.0, and 4.0 in Japanese residential areas. These FAR regulations are determined by the land use purpose. In addition, FAR regulations are determined by the combination of the road width and the land use purpose. This is called reference FAR regulation. In a later section, we determine that 2.0 is an important threshold FAR. In the case of 2.0, if the road is less than 5 m wide, the effective FAR will be determined by the reference FAR regulation. However, we can identify only wide, medium, or narrow for the width, not the exact width. Furthermore, this effect is not so large (e.g., even if the width is 4m, 1.6 is the reference). Therefore, we ignore the effect of the reference FAR regulation.

We explored which FAR is a threshold differentiating the level of benefit of undergrounding utility lines. For this treatment, it can be argued that FAR regulation cannot completely represent the real building sizes. This is true. However, we used this FAR regulation for the following reasons. There are no other appropriate data for representing the real building sizes. Actually, in Japan, one source provides building sizes at each plot. However, the information is not complete. For example, the heights of some buildings of less than or equal to two stories are not given. In addition, some lots have no height data.

Accordingly, we cannot completely match the heights of the buildings and the benefits of undergrounding utility lines.

Compared to the insufficient data on building heights, the data on FAR regulations are complete. As we show later, the estimates using the data on FAR regulations show clear results. These clear results show that real building sizes reflect the regulated FAR. Actually, it is often said that regulations on FAR follow the market equilibrium FAR in practical situations. In addition, we use only the average height of the buildings on a road (not the heights of a specific building) because our data unit is a road. Thus, this setting does not generate a large bias in the estimation of the effect of the average height on the landscape.

The current study further explores the multiplier effects of road width and building height. The dummy variable $\delta_{im}$ represents the combination of road width and building height. The prefecture dummy variable $\delta_{ip}$ represents which prefecture the road is in. This dummy variable can take account of the idiosyncratic character of the prefecture. We assume that error term $\varepsilon_i$ follows $N(0, \sigma^2)$, that is, a normal distribution with mean zero and variance $\sigma^2$. The coefficient $\lambda$ in Equation (11) takes some value between zero and one, which expresses the Box-Cox transformation.

The regression coefficients of Equations (9) and (10) are estimated using the ordinary least squares (OLS). The left-hand side (LHS) of Equation (11) is non-linear with respect to $\lambda$, and the likelihood function of Equation (11) is not necessarily concave with respect to all the coefficients including $\lambda$. Therefore, we take the following simple method, which is similar to the method explained by Kanemoto et al. [29]. First, we gradually increase $\lambda$ from 0.0 to 1.0 in increments of 0.05. Given the respective value of $\lambda$, we maximize the likelihood function to estimate the remaining coefficients, i.e., $\alpha_0$ and $\alpha_k$ ($k = 1, 2, \ldots$ K). Finally, we choose the value of $\lambda$ with the maximum likelihood.

### 3.2. Land Price Data: Road Rating

There are several types of land price information in Japan. The best way for the hedonic approach is to use market transaction prices without any appraisal biases. However, to ensure privacy, the locations of transactions and actual transaction prices are not disclosed. Moreover, the sample size is too small for our analysis because the undergrounding sites are geographically sparse.

The Land Market Value Publication, which has often been used in the hedonic approach literature, is appraised at approximately 25,000 locations and assessed by the land appraisal committee under the MLIT. However, the sample size is too small for our analysis. The third data are Land Appraisal for Fixed Asset Tax for fixed asset tax assessment and assessed by municipal governments. The data are appraised every three years. The sample size is approximately 440,000 locations. The last one is Road Rating (valuation of inheritance tax) for inheritance tax and gift tax assessment and assessed by the national tax agency. The sample size is approximately 340,000 locations.

We used the data of Road Rating (valuation for inheritance tax), not Road Rating (valuation for fixed asset tax). Because the latter is collected by municipal governments, there is a possibility that land appraisal might be biased because the local government has an incentive for over- or under-estimation for various reasons.

These data define a "road" as a road between two intersections, and it is appraised at almost every road in urban areas so that most undergrounding sites can be used for estimation. Since we used the database of the undergrounded lines up to the year 2015, we used the land appraisal prices in 2015, which are obtained from the Research Center for Property Assessment System. Since 1994, the assessments have been mandated to be aimed to produce the values that are 80% of the values of the Land Market Value Publication. Thus, we needed to divide the WTP estimated by 0.8 to adjust the scale and derive the market value.

One possible criticism of the use of appraisal land values is that appraisers may not take into account the economic value of undergrounding utility lines. However, the appraisers typically consider nearby market transactions. Hence, the value of undergrounding

utility lines is likely to be reflected in appraisal values even if an appraiser does not consider the value of undergrounding directly. Indeed, many studies which take the hedonic approach (e.g., Tsutsumi and Seya [30], and Yazawa and Kanemoto [31]) in Japan adopt the use of appraisal value (the Land Market Value Publication).

### 3.3. Study Area and Data on Undergrounded Lines

Our study area was all of Japan. We used the longitude and latitude information of roads on Mapion URL. As a result, we have thirty-three prefectures and 1591 lines that are undergrounded in residential areas, which can be identified with the 2015 Mapion URL. In the case of a small number of undergrounded lines, there is a possibility that the lines might have been undergrounded for specific reasons, such as political reasons. Thus, we chose fifteen further prefectures that have more than twenty undergrounded utility lines. In total, 1464 undergrounded lines are counted in our target prefectures.

We have two types of analyses. In analysis (1), we explored the effects of road width and building height on WTP for undergrounding utility lines, and in analysis (2), we appraised the ratio of costs to benefits (B/C, hereafter) for undergrounded utility lines. Analysis 1 uses the data of the Tokyo metropolitan area in Japan (Saitama, Chiba, Tokyo, Kanagawa). This area has a large number of high buildings so that we could explore the effects of building height. Analysis 2 is for the fifteen prefectures. The fifteen prefectures, including the Tokyo metropolitan area, are Hokkaido, Ibaraki, Tochigi, Saitama, Chiba, Tokyo, Kanagawa, Niigata, Gifu, Aichi, Mie, Kyoto, Osaka, Hyogo, and Fukuoka.

### 3.4. Dependent Variable and Explanatory Variables

The dependent variable $P_i$ in Equations (9)–(11) denotes land price (yen/m$^2$). We made a dummy variable of whether the utility line along the road is undergrounded or not (Yes: 1, No: 0). If only a part of the road has undergrounded lines, we regarded the road as an undergrounded road.

Undergrounding utility lines may have spatial spillover effects. The effects are composed of esthetic benefits, improved traffic flow, and increased safety in disaster situations. From the esthetic viewpoint, residents walk in their neighborhood and therefore might enjoy the roads with undergrounded lines. From the viewpoints of traffic flow as well as safety in disaster situations, residents use the roads with undergrounded lines.

To capture such neighborhood benefits, we defined "the roads in the neighborhood not facing roads with undergrounded lines" as the roads which at least partly lie within 50 m of the roads with undergrounded lines. Note that these neighboring roads do not include the road with undergrounded lines. We call this area "the neighborhood". The definition of the neighborhood is shown in Figure 1.

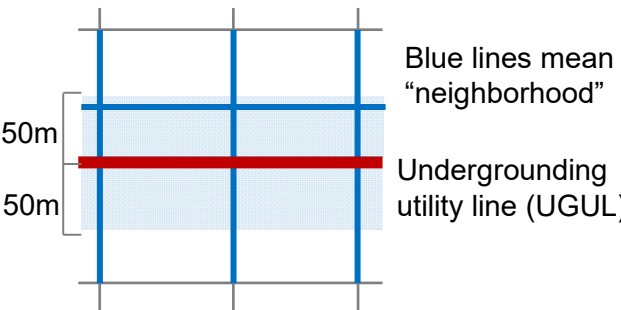

**Figure 1.** Definition of the neighborhood.

We show the number of roads with undergrounded utility lines (abbreviated as UGUL in the tables) in the neighborhood by road width, by area ratio, and by prefecture in Table 1. The number of roads in parentheses is the number of roads with undergrounded utility lines. Table 2 shows the descriptive statistics on land price for analysis 1 and analysis 2 (1000 JPY). We used land price data of the municipalities that have undergrounded lines.

Furthermore, in order to eliminate the effects of outliers, we excluded the top 5% and the bottom 5% of land prices in each prefecture.

**Table 1.** Number of roads in each prefecture.

| Prefectures | Total | Neighborhood | Road Width | | | Floor Area Ratio | | | | | | | |
|---|---|---|---|---|---|---|---|---|---|---|---|---|---|
| | | | Wide | Medium | Narrow | 4 | 3 | 2 | 1.5 | 1 | 0.8 | 0.6 | 0.5 |
| Hokkaido | 78,628 | 210 | 1817 | 21,995 | 54,816 | 0 | 428 | 46,530 | 82 | 2409 | 21,453 | 7726 | 0 |
| | 28 | - | 14 | 9 | 5 | 0 | 0 | 6 | 0 | 0 | 22 | 0 | 0 |
| Ibaraki | 14,826 | 56 | 74 | 2329 | 12,423 | 0 | 104 | 8829 | 760 | 2862 | 2271 | 0 | 0 |
| | 21 | - | 0 | 10 | 11 | 0 | 0 | 16 | 2 | 2 | 1 | 0 | 0 |
| Tochigi | 26,910 | 169 | 174 | 4419 | 22,317 | 0 | 0 | 21,736 | 0 | 160 | 2368 | 2646 | 0 |
| | 38 | - | 0 | 25 | 13 | 0 | 0 | 38 | 0 | 0 | 0 | 0 | 0 |
| Saitama | 45,987 | 484 | 273 | 7006 | 38,708 | 0 | 2 | 31,535 | 1630 | 7292 | 5418 | 110 | 0 |
| | 71 | - | 8 | 42 | 21 | 0 | 0 | 52 | 0 | 1 | 0 | 18 | 0 |
| Chiba | 76,292 | 1032 | 564 | 15,734 | 59,994 | 0 | 90 | 34,498 | 8119 | 30,814 | 2281 | 490 | 0 |
| | 218 | - | 14 | 113 | 91 | 0 | 14 | 172 | 3 | 26 | 3 | 0 | 0 |
| Tokyo | 99,894 | 968 | 784 | 15,474 | 83,636 | 906 | 11,341 | 35,592 | 13,653 | 16,983 | 20,918 | 501 | 0 |
| | 418 | - | 16 | 184 | 218 | 2 | 54 | 214 | 19 | 51 | 72 | 6 | 0 |
| Kanagawa | 72,332 | 350 | 589 | 14,056 | 57,687 | 98 | 11 | 33,269 | 4696 | 10,941 | 22,796 | 521 | 0 |
| | 127 | - | 7 | 56 | 64 | 0 | 0 | 66 | 2 | 14 | 45 | 0 | 0 |
| Niigata | 28,879 | 295 | 105 | 3901 | 24,873 | 0 | 0 | 24,578 | 356 | 3003 | 942 | 0 | 0 |
| | 86 | - | 6 | 43 | 37 | 0 | 0 | 80 | 0 | 4 | 2 | 0 | 0 |
| Gifu | 26,170 | 92 | 233 | 4465 | 21,472 | 0 | 108 | 23,885 | 114 | 651 | 1169 | 243 | 0 |
| | 23 | - | 0 | 14 | 9 | 0 | 1 | 22 | 0 | 0 | 0 | 0 | 0 |
| Aichi | 78,954 | 100 | 481 | 13,553 | 64,920 | 0 | 187 | 57,125 | 8400 | 9049 | 3090 | 1103 | 0 |
| | 27 | - | 4 | 13 | 10 | 0 | 0 | 24 | 2 | 1 | 0 | 0 | 0 |
| Mie | 29,040 | 116 | 407 | 4997 | 23,636 | 0 | 10 | 23,225 | 0 | 2940 | 2865 | 0 | 0 |
| | 26 | - | 0 | 23 | 3 | 0 | 0 | 12 | 0 | 0 | 14 | 0 | 0 |
| Kyoto | 45,578 | 573 | 732 | 8682 | 36,164 | 0 | 1694 | 30,695 | 36 | 2804 | 7866 | 2483 | 0 |
| | 136 | - | 7 | 59 | 70 | 0 | 2 | 75 | 0 | 1 | 34 | 24 | 0 |
| Osaka | 120,646 | 402 | 495 | 16,635 | 103,516 | 61 | 8542 | 93,743 | 3315 | 11,131 | 3854 | 0 | 0 |
| | 74 | - | 3 | 37 | 34 | 1 | 14 | 52 | 0 | 7 | 0 | 0 | 0 |
| Hyogo | 90,548 | 531 | 586 | 13,517 | 76,445 | 5 | 1012 | 56,686 | 7784 | 17,772 | 7289 | 0 | 0 |
| | 106 | - | 31 | 36 | 39 | 0 | 3 | 64 | 9 | 30 | 0 | 0 | 0 |
| Fukuoka | 33,324 | 336 | 521 | 10,195 | 22,608 | 0 | 79 | 23,125 | 2691 | 1281 | 4706 | 1442 | 0 |
| | 65 | - | 11 | 36 | 18 | 0 | 0 | 60 | 0 | 0 | 3 | 2 | 0 |
| (1) Tokyo metropolitan | 294,505 | 2834 | 2210 | 52,270 | 240,025 | 1004 | 11,444 | 134,894 | 28,098 | 66,030 | 51,413 | 1622 | 0 |
| | 834 | - | 45 | 395 | 394 | 2 | 68 | 504 | 24 | 92 | 120 | 24 | 0 |
| (2) Total | 868,008 | 5714 | 7835 | 156,958 | 703,215 | 1070 | 23,608 | 545,051 | 51,636 | 120,092 | 109,286 | 17,265 | 0 |
| | 1464 | - | 121 | 700 | 643 | 3 | 88 | 953 | 37 | 137 | 196 | 50 | 0 |

Note: The number of roads in parentheses is the number of roads with undergrounded utility lines.

**Table 2.** The descriptive statistics on land price.

| Analysis 1 | | (Unit: 1000 JPY) | | | | |
|---|---|---|---|---|---|---|
| UGUL, All or Neighborhood | Variable | Road Width | | | | Total |
| | | Narrow | | Medium/Wide | | |
| | | Building Height | | | | |
| | | (i) Low | (ii) High | (iii) Low | (iv) High | |
| UGUL | Mean | 161 | 188 | 132 | 191 | 178 |
| | Median | 145 | 160 | 115 | 145 | 145 |
| | Standard deviation | 104 | 125 | 84 | 127 | 119 |
| | Sample size | 175 | 219 | 85 | 355 | 834 |
| All lines | Mean | 153 | 140 | 162 | 155 | 156 |
| | Median | 130 | 125 | 130 | 130 | 130 |
| | Standard deviation | 99 | 86 | 107 | 99 | 101 |
| | Sample size | 124,001 | 116,024 | 23,162 | 31,318 | 294,505 |
| Neighborhood | Mean | - | - | - | - | 151 |
| | Median | - | - | - | - | 120 |
| | Standard deviation | - | - | - | - | 107 |
| | Sample size | | | | | 2834 |

**Table 2.** *Cont.*

| Analysis 2 | | | | | |
|---|---|---|---|---|---|
| **UGUL, All or Neighborhood** | **Variable** | **Narrow** | **Road Width Medium** | **Wide** | **Total** |
| UGUL | Mean | 141 | 139 | 98 | 136 |
| | Median | 115 | 110 | 67 | 105 |
| | Standard deviation | 107 | 112 | 59 | 107 |
| | Sample size | 643 | 700 | 121 | 1464 |
| All lines | Mean | 101 | 98 | 98 | 101 |
| | Median | 82 | 81 | 78 | 81 |
| | Standard deviation | 80 | 75 | 75 | 79 |
| | Sample size | 703,215 | 156,958 | 7835 | 868,008 |
| Neighborhood | Mean | - | - | - | 114 |
| | Median | - | - | - | 86 |
| | Standard deviation | - | - | - | 89 |
| | Sample size | | | | 5714 |

Note: UGUL means undergrounded utility lines. In the case of analysis 1, we combined medium width and wide roads, because the number of wide roads is small. High buildings are buildings with a floor area ratio of more than 2.

The list of explanatory variables is shown in Table 3. Table 3 includes the name of the variable, description, year, data source, and expected sign condition. Because we used the land price data for the year 2015, explanatory variables are prepared for the same year if possible and for the nearest available year if not.

**Table 3.** Explanatory variables for our estimation.

| No. | Definition | | Year | Date Resource | Expected Sign |
|---|---|---|---|---|---|
| 1 | Distance to | the main station in the prefecture (m) | 2014 | | − |
| 2 | | the nearest station (m) | 2014 | | − |
| 3 | | the nearest bus station (m) | 2010 | | − |
| 4 | | the nearest primary school (m) | 2013 | | − |
| 5 | | the nearest middle school (m) | 2013 | | − |
| 6 | | the nearest hospital (m) | 2014 | National Land Numerical Information (No.1–15) | − |
| 7 | | the nearest post office (m) | 2013 | | − |
| 8 | | the nearest clinic (m) | 2014 | | − |
| 9 | Maximum floor area ratio (%) | | 2011 | | +,− |
| 10 | Maximum building coverage ratio (%) | | 2011 | | +,− |
| 11 | Each of the following seven residential districts are considered as a dummy variable (i.e., 1 or 0). (i) Category 1 low-rise exclusive residential districts (ii) Category 2 low-rise exclusive residential districts (iii) Category 1 medium-to-high-rise exclusive residential districts (iv) Category 2 medium-to-high-rise exclusive residential districts (v) Category 1 residential districts (vi) Category 2 residential districts (vii) Quasi-residential districts | | 2011 | | |
| 12 | Landscape planning area, 1; other, 0 | | 2014 | | + |
| 13 | Landscape emphasis planning area, 1; other, 1 | | 2014 | | + |
| 14 | Distance from the center of a newtown is less than 460 m, 1; others, 0 | | 2015 | | + |
| 15 | Distance from the center of a newtown developed before 1976 is less than 460 m, 1; others, 1 | | 2014 | | − |
| 16 | Dummy variable on the width of a road whether narrow or medium or wide | | 2015 | DRM-DB | − |
| 17 | Dummy representing each municipality | | 2015 | - | − |
| 18 | Undergrounded utility line, 1; others, 0 | | 2015 | - | + |
| 19 | Areas within 50 m of undergrounded utility lines, 1; others, 0 | | 2015 | - | + |
| 20 | Dummy variable for the width of a road whether narrow (1) or wide (0) | | 2015 | DRM-DB | + |
| 21 | Dummy variable for the height of buildings whether floor area ratio is more than 2, 1; others, 0 | | 2011 | -National Land Numerical Information | − |
| 22 | Dummy representing each prefecture | | 2015 | - | − |

Note: DRM-DB implies Database of Digital Road Map.

We discuss the meaning and the expected signs of the explanatory variables one by one as follows. The numbers in parentheses in the head of each paragraph correspond to the numbers in the first column in Table 3. The explanatory variables that are strongly related are explained together.

(1)–(8) The Euclidian distances from the main station, the nearest station, the nearest bus station, the nearest elementary school, the nearest junior high school, the nearest hospital, the nearest post office, and nearest clinic are used as explanatory variables. We assume that the signs for these variables are negative. One station in each prefecture is chosen as the main station in terms of the number of passengers and the connection to Shinkansen stations. The chosen stations are Sapporo, Sendai, Mito, Utsunomiya, Omiya, Chiba, Tokyo, Yokohama, Nigata, Gifu, Nagoya, Tsu, Kyoto, Osaka, Sannomiya, and Hakata.

(9), (10) Regulation value of the floor area ratios and regulation value of the building coverage ratios are used as explanatory variables. When the regulation value of floor area ratio is high, the land price can increase because landowners can build a high building. On the other hand, buildings surrounding the land can also be high because the floor area ratio is common in a certain area. High buildings can block sunlight to the building, thereby decreasing the land prices. Regarding the regulation value of the building coverage ratio, when this ratio is high, land prices can increase because landowners can build large buildings. However, an area with large buildings is unfavorable from the viewpoint of the landscape. Accordingly, the signs for the floor area ratio and the building coverage ratio can be either positive or negative.

(11) Types of restrictions on land use are used as explanatory variables. To account for heterogenous impacts of different restrictions on land use, we considered seven types of restrictions on land use that are imposed in residential areas. We show the seven land use districts in Table 3.

(12), (13) Esthetic landscapes can increase land prices, so we use "landscape planning area" and "landscape emphasis area" as dummy variables. A landscape planning area has restrictions on building heights and designs to protect esthetic landscapes. A landscape emphasis area, which is set within a landscape planning area, has a unique landscape and should be well protected.

(14), (15) Newly developed residential areas can provide good environments. Thus, the land prices tend to be higher than in other areas. On the other hand, areas which were developed a long time ago normally have a high proportion of elderly residents and many vacant houses, which have negative effects. To account for these effects, we set dummies for newly developed areas and old developed areas. We defined the areas developed before 1976 as the "older developed areas", whereas the areas developed after 1976 are defined as the "newly developed areas". However, it is hard to know how large each newly developed residential area is. Thus, since the median value of Japanese developed residential areas is 67 ha, we assume that each developed area has a 460 m radius. A circle with this radius has an area of about 67 ha.

(16) The road width dummy variable is used as an explanatory variable. It defines whether road width is narrow (less than 5.5 m), medium (between 5.5 and 13 m), or wide (more than 13 m).

(17) Municipality dummy variables are used as explanatory variables in order to take account of municipality-specific factors which cannot be explained by other explanatory variables.

(18), (19), (20), (21), (22) Dummies representing whether the utility lines on the road are undergrounded or not, whether the road is in the neighborhood of the road with undergrounded lines, whether the road is narrow or wide, whether the floor area ratio is more than 2 or not, and where the road is, are used as explanatory variables. Note that we already define the neighborhood in the third paragraph of Section 3.4. We have already defined these variables in detail in Section 3.1.

Finally, projects of undergrounding utility lines often involve refurbishing roads and sidewalks, widening sidewalks, and planting street trees. Unfortunately, it is hard to collect information on such projects associated with undergrounding projects. Hence, the measured benefit includes these benefits too.

## 4. Empirical Analysis

### 4.1. Parameter Estimation

This section shows the regression coefficients for analysis 1 and analysis 2. We used three types of functional forms (linear, full log, semi Box-Cox). For the Box-Cox type (Equation (11)), in all estimations, the values of $\lambda$ which maximize the likelihood are estimated to be zero. That implies that the LHS of Equation (11) is $\ln(P)$ because $\lim_{\lambda \to 0} (p^\lambda - 1)/\lambda = \ln p$.

We show the regression coefficients related to undergrounded utility lines and the $p$ values for analysis 1 in the three functional forms in Table 4. We explored which ratio is a threshold differentiating the level of benefit of undergrounding utility lines. The regulation values of floor area ratios 3, 2, and 1.5 are used as candidates for the threshold. The estimated parameters related to undergrounded utility lines are classified into two: (1) UGUL and (2) Neighborhood. (1) UGUL means the parameter estimated for the roads with undergrounded utility lines. (2) Neighborhood means the parameter estimated for the road in the neighborhood of undergrounded utility lines. The parameter "Basis" in (1) UGUL implies the parameter for narrow roads with low buildings. The parameters "Medium/Wide" and "High" represent the parameters for undergrounded lines with medium or wide roads and high buildings. The parameter "Medium/Wide and High" captures the multiplier effects of road width and building height. Therefore, the combination of the parameters can represent the change in land price in any kind of road.

Regarding the dummy variables for the prefecture, we chose the regression result in which all $p$ values of the dummy variable for prefecture were greater than 0.05. In this process, we adopted the backward stepwise method. We do not show the regression coefficients of the other explanatory variables because it is unnecessary for our analysis. Those coefficients are shown in the Supplement, which can be obtained from the authors.

In analysis 1, we chose the regression results in which the threshold for the floor area ratio is 2 and the functional form is semi-log for the following reason. Regarding the threshold, in all the regression forms, R squared is common regardless of the thresholds. This is because many roads have no undergrounded lines, so the difference in the threshold does not affect R squared. However, AIC is affected slightly. In the linear form and the semi-log form, AIC is minimized in the case of the threshold of 2.0. In the full-log form, AIC is minimized in the case of the threshold of 3.0. Correspondingly, in the regression results in which the threshold floor area ratio is 2, the $p$ value of the dummy variable for building height (i.e., High) is lower than the other thresholds 3 and 1.5 and is less than 0.05.

In terms of the functional forms, judging from R squared, the semi-log form is the best. Actually, judging from the $p$ value of the dummy variable of building height, the full-log functional form is not reliable. The Box-Cox type (Equation (11)) can represent the linear type and the semi-log type. The value of $\lambda$ which maximizes the likelihood was estimated to be zero (i.e., the semi-log type), so we adopted the regression coefficient in the case of $\lambda = 0$.

**Table 4.** Regression coefficients for the UGUL in (analysis 1).

**Linear Functional Form**

| Variables | FAR Threshold of 3 | | | FAR Threshold of 2 | | | FAR Threshold of 1.5 | | |
|---|---|---|---|---|---|---|---|---|---|
| | Coefficient | *p* Value | | Coefficient | *p* Value | | Coefficient | *p* Value | |
| Intercept | 144,831 | 0 | *** | 144,819 | 0 | *** | 144,848 | 0 | *** |
| (1) UGUL | | | | | | | | | |
| Basis | 28,526 | $6.71 \times 10^{-28}$ | *** | 33,459 | $8.13 \times 10^{-31}$ | *** | 30,345 | $5.99 \times 10^{-25}$ | *** |
| Medium/Wide | −8248 | $3.44 \times 10^{-5}$ | *** | −20,849 | $9.08 \times 10^{-9}$ | *** | −19,126 | $5.25 \times 10^{-7}$ | *** |
| High | −11,732 | $6.78 \times 10^{-2}$ | | −10,907 | $1.14 \times 10^{-4}$ | *** | −3571 | $2.12 \times 10^{-1}$ | |
| Medium/Wide and High | 16458 | $2.92 \times 10^{-2}$ | ** | 20,192 | $2.68 \times 10^{-6}$ | *** | 15,490 | $4.67 \times 10^{-4}$ | *** |
| (2) Neighborhood | 5558 | $8.03 \times 10^{-27}$ | *** | 5557 | $8.20 \times 10^{-27}$ | *** | 5556 | $8.29 \times 10^{-27}$ | *** |
| Sample size | 294,504 | | | 294,504 | | | 294,504 | | |
| R-squared | 0.929 | | | 0.929 | | | 0.929 | | |
| AIC | 6,847,407 | | | 6,847,389 | | | 6,847,398 | | |
| *p* value of F-Statistic < | $2.20 \times 10^{-16}$ | | | $2.20 \times 10^{-16}$ | | | $2.20 \times 10^{-16}$ | | |

**Semi-log functional form**

| Variables | FAR Threshold of 3 | | | FAR Threshold of 2 | | | FAR Threshold of 1.5 | | |
|---|---|---|---|---|---|---|---|---|---|
| | Coefficient | *p* Value | | Coefficient | *p* Value | | Coefficient | *p* Value | |
| Intercept | 11.92 | 0 | *** | 11.92 | 0 | *** | 11.92 | 0 | *** |
| (1) UGUL | | | | | | | | | |
| Basis | 0.2032 | $2.03 \times 10^{-80}$ | *** | 0.2320 | $2.58 \times 10^{-57}$ | *** | 0.2173 | $6.32 \times 10^{-48}$ | *** |
| Medium/Wide | −0.0760 | $4.02 \times 10^{-10}$ | *** | −0.1130 | $3.05 \times 10^{-7}$ | *** | −0.1051 | $5.69 \times 10^{-6}$ | *** |
| High | −0.0614 | $1.19 \times 10^{-1}$ | | −0.0515 | $2.53 \times 10^{-3}$ | ** | −0.0251 | $1.46 \times 10^{-1}$ | |
| Medium/Wide and High | 0.0466 | $3.14 \times 10^{-1}$ | | 0.0627 | $1.70 \times 10^{-2}$ | * | 0.0433 | $1.08 \times 10^{-1}$ | |
| (2) Neighborhood | 0.0590 | $7.94 \times 10^{-77}$ | *** | 0.0590 | $8.83 \times 10^{-77}$ | *** | 0.0590 | $8.39 \times 10^{-77}$ | *** |
| Sample size | 294,504 | | | 294,504 | | | 294,504 | | |
| R-squared | 0.932 | | | 0.932 | | | 0.932 | | |
| AIC | −221,844 | | | −221,851 | | | −221,845 | | |
| *p* value of F-Statistic < | $2.20 \times 10^{-16}$ | | | $2.20 \times 10^{-16}$ | | | $2.20 \times 10^{-16}$ | | |

**Full-log functional form**

| Variables | FAR Threshold of 3 | | | FAR Threshold of 2 | | | FAR Threshold of 1.5 | | |
|---|---|---|---|---|---|---|---|---|---|
| | Coefficient | *p* Value | | Coefficient | *p* Value | | Coefficient | *p* Value | |
| Intercept | 15.10 | 0 | *** | 15.10 | 0 | *** | 15.10 | 0 | *** |
| (1) UGUL | | | | | | | | | |
| Basis | 0.1213 | $3.33 \times 10^{-18}$ | *** | 0.1212 | $2.35 \times 10^{-13}$ | *** | 0.1041 | $6.95 \times 10^{-10}$ | *** |
| Medium/Wide | −0.0382 | $2.00 \times 10^{-3}$ | ** | −0.0681 | $2.43 \times 10^{-3}$ | ** | −0.0581 | $1.37 \times 10^{-2}$ | * |
| High | −0.0456 | $2.54 \times 10^{-1}$ | | 0.0052 | $7.67 \times 10^{-1}$ | | 0.0380 | $3.18 \times 10^{-2}$ | * |
| Medium/Wide and High | 0.0245 | $6.02 \times 10^{-1}$ | | 0.0351 | $1.88 \times 10^{-1}$ | | 0.0136 | $6.20 \times 10^{-1}$ | |
| (2) Neighborhood | 0.0632 | $3.73 \times 10^{-85}$ | *** | 0.0632 | $3.93 \times 10^{-85}$ | *** | 0.0632 | $3.99 \times 10^{-85}$ | *** |
| Sample size | 294,504 | | | 294,504 | | | 294,504 | | |
| R-squared | 0.930 | | | 0.930 | | | 0.930 | | |
| AIC | −212,304 | | | −212,306 | | | −212,312 | | |
| *p* value of F-Statistic < | $2.20 \times 10^{-16}$ | | | $2.20 \times 10^{-16}$ | | | $2.20 \times 10^{-16}$ | | |

Note: * significant at 5% level, ** significant at 1% level, and *** significant at 0.1% level. 1. UGUL means undergrounded utility lines. 2. We combined medium width and wide roads into one category because the number of wide roads is small. 3. Medium/Wide means medium width and wide roads. 4. High means high buildings. 5. Medium/Wide and High means medium width and wide road and high buildings.

Next, we show the regression coefficients related to undergrounded utility lines and the *p* values in all functional forms for analysis 2 in Table 5. The parameter "Basis" in (1) UGUL implies the parameter for narrow roads. The parameters "Medium" and "Wide" represent the parameters for undergrounded lines with medium width and wide roads. These coefficients are shown in the Supplement, which can be obtained from the authors. For the analysis that appraises the benefit–cost ratio (B/C) for undergrounded utility lines, we used the coefficients of all functional forms because the *p* value is low in all variables and all functional forms.

In Section 4.2, we estimate WTP for undergrounding utility lines. Before that, we compare our estimated parameters with those of previous papers. In terms of the rate of change in the property value, we can compare our results to Oba [4], which targets the Kyoto undergrounding projects. However, analysis 1 does not include Kyoto areas. Using our parameters of analysis 2, we can calculate the rate of increase in land prices in Kyoto as 14–25% depending on the width of roads. Oba's results show that the rate of increase is 21.9%. As such, our estimates are close to his value.

**Table 5.** Regression coefficients for the UGUL in all functional forms (analysis 2).

| Variables | Linear | | | Semi-Log | | | Full-Log | | |
|---|---|---|---|---|---|---|---|---|---|
| | Coefficient | *p* Value | | Coefficient | *p* Value | | Coefficient | *p* Value | |
| Intercept | 63,560 | 0 | *** | 11.01 | 0 | *** | 13.61 | 0 | *** |
| (1) UGUL | | | | | | | | | |
| Basis | 17,343 | $2.23 \times 10^{-38}$ | *** | 0.1964 | $4.68 \times 10^{-79}$ | *** | 0.0896 | $2.77 \times 10^{-15}$ | *** |
| Medium | −2561 | $2.57 \times 10^{-2}$ | * | −0.0603 | $4.25 \times 10^{-9}$ | *** | −0.0524 | $4.52 \times 10^{-7}$ | *** |
| Wide | −7678 | $4.60 \times 10^{-4}$ | *** | −0.1144 | $2.01 \times 10^{-9}$ | *** | −0.1125 | $6.66 \times 10^{-9}$ | *** |
| (2) Neighborhood | 8184 | $1.39 \times 10^{-184}$ | *** | 0.0892 | $8.04 \times 10^{-276}$ | *** | 0.0865 | $1.36 \times 10^{-253}$ | *** |
| Sample size | 868,007 | | | 868,007 | | | 868,007 | | |
| R-squared | 0.931 | | | 0.937 | | | 0.936 | | |
| AIC | 19,723,724 | | | -465,377 | | | −45,025 | | |
| *p* value of F-Statistic | < $2.20 \times 10^{-16}$ | | | $2.20 \times 10^{-16}$ | | | $2.20 \times 10^{-16}$ | | |

Note: 1. * significant at 5% level, and *** significant at 0.1% level. UGUL means undergrounded utility lines. 2. Medium means a medium width road. 3. Wide means a wide road.

We can compare our results with those in Canada, the UK, and Australia. In Canada, studies have shown losses in the value of properties adjacent to an overhead line of 6.3% (Hamilton and Schwann [7]) and 9.6% on average (François [9]). In the UK, the values of the property within 100 m of an overhead line are reduced by 6–17% (an average of 11.5%) (Sims and Dent [8]). In Australia, the values of property with an overhead line are reduced by 2.9% (McNair and Abelson [5]). Returning to our results in Japan, in our full-log case in analysis 2, the base parameter for undergrounding is estimated at 0.0896, which implies that the property value with overhead utility lines is reduced by about 9% along narrow roads in Japan. Actually, about 80% of our target roads (as shown in Table 1) are classified as narrow roads. Comparing the cases in Japan with those in Canada, the UK, and Australia, the effects of underground utility lines on property values are similar in Japan. However, the effects depend on the width of the roads. Indeed, the estimated dummy for medium width roads in our full-log case in analysis 2 is −0.0524, which implies that the property value with overhead utility lines is reduced by about 3.7% (=100(0.0896 − 0.0524)) along medium width roads.

### 4.2. Analysis

#### 4.2.1. Analysis 1: Exploring the Effects of Road width and Building Height

The WTP for undergrounding utility lines can be calculated with the increase in the land price associated with undergrounded lines and the average housing lot size. The increase in land price associated with undergrounding lines, $\Delta P = dP/dz_{ix}$, in front of the house and in the neighborhood, is calculated according to the specified functions, as shown in the equations $\Delta P = \alpha_x$ (linear type), and $\Delta P = (e^{\alpha_x} - 1)P^*/e^{\alpha_x}$ (semi-log and full-log types), where $\alpha_x$ is the regression coefficient for the *x*th attribute from Equations (9)–(12).

The estimated values of the WTP for undergrounding projects are summarized in the four categories shown as (i), (ii), (iii), and (iv) in Table 6 and Figure 2. The four categories are classified by the combination of road width and building height. WTP can be calculated by multiplying the increase in the land price by the lot area according to Equation (8). Therefore, WTP depends on the land price and lot size. However, we calculate the WTP by using the building area, not by the lot size, due to data constraints. The lot size includes areas other than the building area, such as a garden. Thus, the exact value of WTP shown in Table 6 and Figure 2 would increase. To calculate these values, the evaluation ratio of the land rating to real land prices, 0.8, is already taken into account. We used the median value of the data along the road classified in the respective category. Since we used the prefecture dummy of Tokyo, we needed to use this coefficient additionally to calculate the WTP in Tokyo. Because the current analysis is concerned with the determinants of the WTP for undergrounding lines, we focused only on the WTP in Saitama, Chiba, and Kanagawa. The WTP in Tokyo can be calculated just by adding some values to the calculated WTP in these three prefectures.

**Table 6.** Calculated WTP for UGUL.

| | | Road Width | | | |
| --- | --- | --- | --- | --- | --- |
| | | Narrow | | Medium/Wide | |
| | | Building Height | | | |
| | | (i) Low | (ii) High | (iii) Low | (iv) High |
| (1) Regression coefficient | | 0.232 | 0.181 | 0.119 | 0.130 |
| (2) Median value of floor area | ($m^2$/household) | 59 | 55 | 64 | 58 |
| (3) Median value of land price | (1000 JPY) | 118 | 124 | 125 | 131 |
| (4) Benefits | (1000 JPY/$m^2$) | 27 | 22 | 15 | 17 |
| (5) WTP | (1000 JPY) | 1614 | 1225 | 958 | 997 |

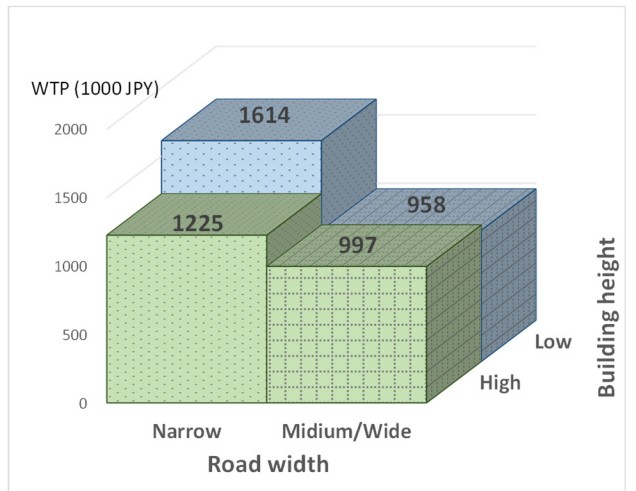
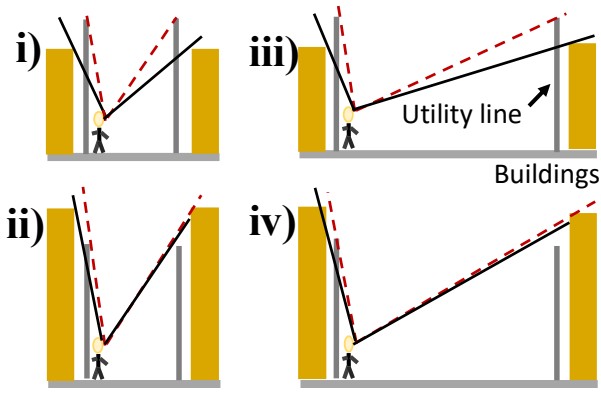

**Figure 2.** WTPs and images of the increase in the visible expanse of the sky. Note: (i), (ii), (iii), and (iv) correspond to the four categories in Table 6. The dashed lines imply the range of sky view without utility lines; the solid lines represent the range of sky view with utility lines.

From Table 6 and Figure 2, the main characteristics of the results are summarized as Property 1.

Property 1.

(1) The WTP is higher as road width becomes narrower;
(2) The WTP is higher as building height becomes lower;
(3) In the case of wide roads, there is very little difference between the WTP with high buildings and that with low buildings.

We conjecture that these properties depend on the increase in the visible expanse of the sky if the utility lines were undergrounded. The images of the increase in the visible expanse of the sky are shown in Figure 2. The dashed lines imply the range of sky view without utility lines and the solid lines represent the range of sky view with utility lines.

Regarding Property 1 (1), in the case of a narrow road, the increase in the visible expanse of the sky is wider than before the utility lines are undergrounded, as shown in Figure 2. On the other hand, in the case of a wide road, the increase in the visible expanse of the sky does not change much whether the utility lines are undergrounded or not.

In respect to Property 1 (2), in the case of low buildings along roads with undergrounded lines, the increase in the expanse of the sky is wider than in the case of high buildings. In addition, the view from a residence's window is improved by undergrounding utility lines in the case that the room is below the level of utility lines. However, the view will not change if the room is above the utility lines.

Regarding Property 1 (3), in the case of a wide road, the increase in the visible expanse of the sky does not change much with the building height. The increase in the expanse of

the sky is wide in the first place. Therefore, the WTP does not change significantly whether buildings are high or low.

Lastly, we compared the levels of our estimates with McNair and Abelson's [5] estimates using the data of Canberra. Their estimate, per household in terms of stock value, is $12,350 (=1.30 million yen with the 2014 average exchange rate of 106 yen/$). Our estimates shown in Table 6 are about 1.61, 1.23, 9.6, and 10.0 million yen depending on the road widths and the building heights. These values are very similar to McNair and Abelson's estimation. However, to calculate our WTP, we used building sizes instead of lot sizes. The lot size is larger than the building area. Thus, if we used lot sizes, our estimates would be larger. In addition, the percentage of the field of view occupied by electric wires in Australia must be far less than in Japan. Considering these conditions, the WTP per household is similar between our estimates and McNair and Abelson's estimates, particularly when we adopted our estimates in the case of wide roads and low buildings. McNair et al. [6] measured the residents' WTP with contingent valuation methods. As they state in the conclusion of their paper, the level of the WTP they estimated was similar to that of the hedonic approach by McNair and Abelson [5]. This means that the estimates by McNair et al. [6] are also similar to ours.

4.2.2. Analysis 2: Appraising Benefit–Cost Ratio (B/C)

This section appraises the benefit–cost ratio in each road with undergrounded utility lines in consideration of the effects of road width. We could calculate the increase in land price in each road by Equations (5)–(7). Since we had the data containing the lengths of the road, we were able to appraise benefits in each road in which lines were undergrounded before 2015 by multiplying the increase in land price by the length of the road.

In order to calculate the benefits of undergrounded utility lines, we defined fifteen meters on both sides of the undergrounded utility lines as the range that is directly affected by the undergrounding because the land facing undergrounded utility lines is directly affected. The average value of the lot sizes in the residences of our analysis area is 230 square meters. If we suppose that the land is square, the side length of the land is about 15 m.

Regarding the range of the neighborhood effects of undergrounding, we define 35 m from the edge of the directly affected area (i.e., 15 m from the road with undergrounded lines) defined above. The neighborhood dummy variable represents the roads that, at least partly, lie within 50 m of the roads with undergrounded lines.

In terms of the semi-log and full-log types of functional forms, the land price of the neighborhood area is required to calculate the benefits (Equations (6) and (7)). We used the weighted average value of the length and the land price of the neighborhood roads. We used the regression coefficients shown in Table 5 to calculate the benefits. In addition, we supposed that the cost is 530,000 yen per meter, based on the MLIT data.

Figure 3 shows the histogram of B/C by functional form. The average value of B/C is calculated to be 2.27 to 2.65, depending on the functional form. This means that the previous undergrounding projects were efficient overall. However, as shown in Figure 3, the B/C takes a wide range of values; some roads have high B/C but 3–17% of roads have less than 1, depending on the functional form. This implies that the utility lines were undergrounded on roads that should not have been undergrounded. In particular, in the case of wide roads, there is a high possibility that the undergrounding project has a smaller

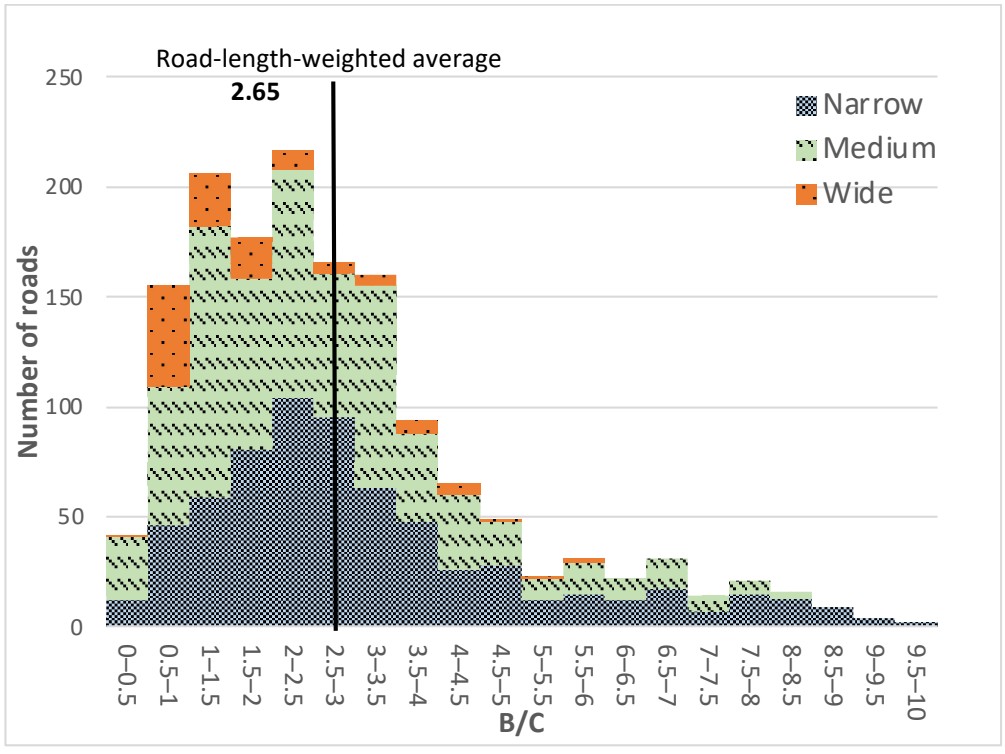

Semi-log type case

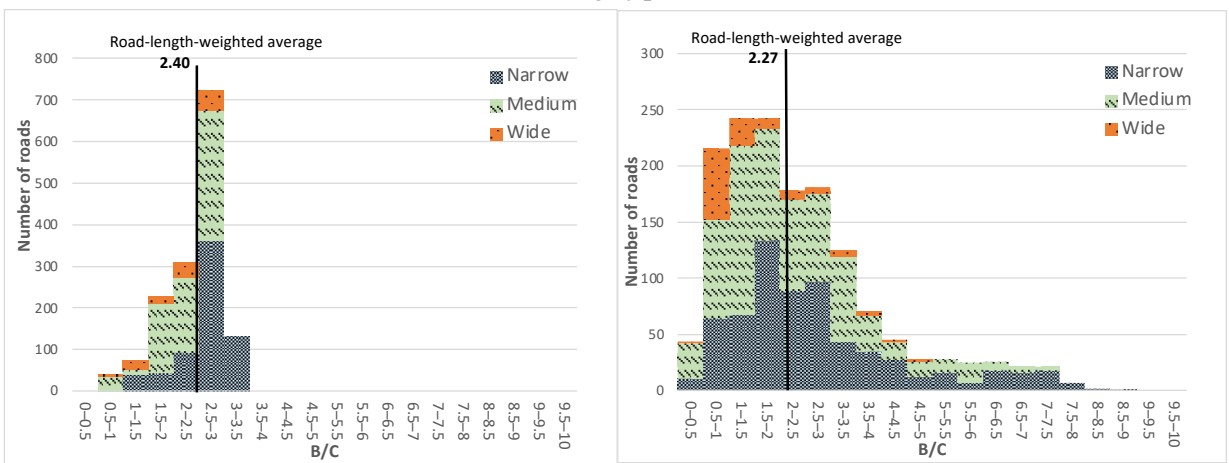

Linear type case

Full-log type case

**Figure 3.** Histogram of B/C in the three functional forms.

### 4.3. Possible Biases of Our Estimation

We used the hedonic approach to estimate the willingness to pay for undergrounding utility lines. The hedonic approach is a typical benefit-measurement method using revealed preference data. In contrast, there are several methods that use stated preference data. In general, in cost–benefit analyses, the measured benefits using the revealed preference data are more reliable than the benefits using the stated preference data because the stated preference data can have relatively large biases. However, the hedonic approach also can have biases in parameter estimation. We will discuss several possible biases in our context.

First, we have discussed reverse causality. The undergrounding of utility lines might be carried out at locations with high land prices. That is, because of the good environment, residents might want utility lines to be undergrounded. However, as long as we measure the benefit using the hedonic approach, this reverse causality has no problem. Regardless of the reasons for undergrounding projects, when lines are undergrounded at a certain

location, the land prices around the location increase. Indeed, Kanemoto [19] proves that perfect mobility between different areas ensures that property prices reflect the benefits of improved amenities and calls this property 'capitalization'. The increases in land price can be captured by land price functions as long as they appropriately control for relevant explanatory variables. This is an advantage of the hedonic approach.

The sites with undergrounded utility poles have not been chosen randomly. As is often noted, samples chosen non-randomly generate several estimation biases (see Baum-Snow et al. [32]). In reality, locations with large benefits from undergrounding may have been selected. As a result, selection biases may occur. In addition, if we cannot capture the amenities generating the high benefits of undergrounding (e.g., good scenery), parameter distortions due to omitted variables may occur. We will explain these in order.

As for the selection biases, there is a possibility that the selection of undergrounding sites might result in an overestimation of our parameters for building heights and road widths. However, there is another possibility that these parameters may not be affected by the selection under the condition that the selected undergrounding sites are similar. If the target sites are similar, the parameters explaining the difference in benefits of undergrounding are estimated using similar samples. In this case, the parameters for building heights and road widths are not overestimated.

As for the omitted variables, our model captures the impact on the benefits of undergrounding as a function of the building height, road width, and prefecture dummies, using Equation (12). In this case, if the omitted variables affecting the benefits of undergrounding are correlated with building height and road width, the parameter estimates in Equation (12) will not be consistent estimators. An example of such omitted variables could be the inherent landscape. In the case of a district with a good landscape, the floor-area ratio of buildings may be kept low, and the setting of road widths may be influenced by the landscape. However, it is almost impossible for us to account for the landscape of each location, or to express the landscape as a quantitative variable and incorporate it into the explanatory variables. As it is difficult to account for the landscape of each location, it is also difficult to propose appropriate instrumental variables. As a result, this problem cannot be addressed in cross-sectional analyses. If time-series data can be obtained, fixed-effect models may be useful to control for region-specific landscapes that do not change over time. This is an issue for future studies.

## 5. Concluding Remarks

In this study, we appraised the increase in housing prices due to the undergrounding of utility lines in Japan and clarify the dependency of the residents' WTP for it on the road width and building height. Our results show that housing prices, as well as the WTP, are lower as road width becomes wider and building height becomes higher. However, when the road is wide, the WTP does not depend on building height. These results can be used to design undergrounding policy guidelines. Specifically, when constructing a cost-benefit manual for undergrounding utility lines, the road widths and the building heights should be taken into account.

The average B/C value of the projects of undergrounding utility lines is from 2.27 to 2.65, depending on the functional forms. Thus, the previous undergrounding projects were efficient on the whole. However, when we look at the B/C ratio of each project, although some roads have high B/C, 3–17% of undergrounding projects have a B/C of less than 1. This suggests that we should conduct cost-benefit analyses for future undergrounding projects.

**Supplementary Materials:** The following are available online at https://www.mdpi.com/article/10.3390/su132414023/s1.

**Author Contributions:** S.I. conducted the analysis and wrote the manuscript. T.K. constructed a methodology and wrote the manuscript. H.S. constructed a methodology and wrote the manuscript. All authors have read and agreed to the published version of the manuscript.

**Funding:** We are grateful to grants from the Ministry of Education, Culture, Sports, Science and Technology, Government of Japan (Grant-in-Aid for Scientific Research 20H01486) and from the Japan Research Center for Transport Policy.

**Institutional Review Board Statement:** Not applicable.

**Informed Consent Statement:** Not applicable.

**Data Availability Statement:** Not applicable.

**Acknowledgments:** We would like to thank three anonymous reviewers for valuable comments on our paper. We are grateful to grants from the Ministry of Education, Culture, Sports, Science and Technology, Government of Japan (Grant-in-Aid for Scientific Research 20H01486) and from the Japan Research Center for Transport Policy. Despite assistance from many sources, any errors in the paper remain the sole responsibility of the authors.

**Conflicts of Interest:** The authors declare no conflict of interest.

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
