# Peer review of "Urban Esthetic Benefits of Undergrounding Utility Lines in Consideration of the Three-Dimensional Landscape"

_sustainability, doi:10.3390/su132414023_

Round 1

Reviewer 1 Report

The present paper quantifies the effect of undergrounding utility lines in Japan through hedonic pricing method. The authors use the land price data of road rating valuation for inheritance tax from the year 2015. Using this one-time cross-sectional data from 47 prefectures in Japan, the paper tries to identify the difference in the appraised land price due to undergrounding utility lines. It shows that the added land values lower when the road becomes wider and the building along the road becomes taller.

The overall quality of the manuscript is fine and easy to follow, except for the lengthy presentation of the estimated results (perhaps due to three alternative estimation models with different functional forms.) The originality and contribution of the paper is that they estimate the heterogeneous effects on the land price by adding controls for different road width and building height (limits) along the road.

My major concern with this study is the establishing the causality. Authors only briefly address the issue of reverse causality in Section 4.3. as a possible bias of estimated coefficients. Although they claim that “as long as we measure the benefit using the hedonic approach, this reverse causality has no problem,” this issue needs to be taken seriously. Since authors present their study as the benefits of undergrounding utility lines, it needs to be clarified that the appraised price premium is that compared to the land that otherwise have similar condition, but their utility lines remain above ground. Although the selection of the road on which utility lines are moved underground should not be random, the present study does not address how the selection is implemented. If there are some attributes which is likely to enhance the probability of undergrounding utility lines (such as low flood risk), there may be the issue of omitted variable bias in addition to the reverse causality.

Discussion on how the selection occurs need to be provided. For the treatment of causality in urban and regional economics (including hedonic pricing method), the following paper should be helpful.

Baum-Snow, Nathaniel, and Fernando Ferreira. "Causal inference in urban and regional economics." Handbook of regional and urban economics. Vol. 5. Elsevier, 2015. 3-68.

Due to data limitation, it seems difficult to establish the causality in the estimation model. At least the existence of these problem, how the authors address the issue, and the limitation of the estimated results should be discussed in more detail.

Author Response

Thank you very much for your valuable comment.  To respond to the main concern “the establishing of the causality”, we have added the following paragraphs to Section 4.3.

“The sites with undergrounded utility poles have not been chosen randomly. As is often noted, the samples chosen non-randomly generate several estimation biases (see Baum-Snow et al., 2015). In reality, locations with large benefits from undergrounding may have been selected. As a result, selection biases may occur. In addition, if we cannot capture the amenities generating the high benefits of undergrounding (e.g., good scenery), parameter distortions due to omitted variables may occur. We explain these in order.

As for the selection biases, there is a possibility that the selection of undergrounding sites might result in an overestimation of our parameters for building heights and road widths. But there is another possibility that these parameters may not be affected by the selection under the condition that the selected undergrounding sites are similar. If the target sites are similar, the parameters explaining the difference in benefits of undergrounding are estimated using the similar samples. In this case, the parameters for building heights and road widths are not overestimated.

As for the omitted variables, our model captures the impact on the benefits of undergrounding as a function of the building height, road width, and prefecture dummies, using eq. (12). In this case, if the omitted variables affecting the benefits of undergrounding are correlated with building height and road width, the parameter estimates in eq. (12) will not be consistent estimators. An example of such omitted variables could be the inherent landscape. In the case of a district with a good landscape, the floor-area ratio of buildings may be kept low, and the setting of road widths may be influenced by the landscape. However, it is almost impossible for us to grasp the landscape of each location, or to express the landscape as a quantitative variable and incorporate it into the explanatory variables. As it is difficult to grasp the landscape of each location, it is also difficult to propose appropriate instrumental variables. As a result, this problem cannot be addressed in cross-sectional analyses. If time-series data can be obtained, fixed effect models may be useful to control for region-specific landscapes that do not change over time. This is an issue for the future.”

Thank you.

Author Response

Reviewer's comments are denoted below and in blue, and our responses are put after [Response].

A.1. Could there have been any possibility that this paper estimates the WTP for other things, not for undergrounding utility lines? The roads for undergrounding utility lines are not randomly determined. The central and local governments have prioritized which utility lines will be underground. Thus, the roads with underground utility lines could have common features. For example, they could be the roads in the political, economic, and cultural central areas or around major stations with many users. Using appropriate control variables (“relevant explanatory variables” in this paper) is crucial to estimate the WTP for undergrounding utility lines. Although the authors use some control variables, they skip the detailed explanation about them. I think they need to clarify what each variable controls in the main text, not in the Supplement (I agree with the authors about skipping the description about estimation results). Otherwise, it remains unconvincing whether the authors indeed estimate the WTP for undergrounding utility lines.

[Response]

As the reviewer points out, the sites with undergrounded utility lines have not been chosen randomly. So we need to explain what independent variables explain. To respond to this comment, we first have moved the part which explains the meaning of each variable, which was relegated to our supplement in the original version, to the main text. In addition, we believe that the discussion in Section 4.3, which was revised extensively in the first-round revision, is useful for this comment.

The text moved from the supplement is “We discuss the meaning and the expected signs of the explanatory variables one by one as follows. The numbers in parentheses in the head of each paragraph correspond to the numbers in the first column in Table 3. The explanatory variables which are strongly related are explained together.

(1)– (8) The Euclidian distances from the main station, the nearest station, the nearest bus station, the nearest elementary school, the nearest junior high school, the nearest hospital, the nearest post office, and nearest clinic are used as explanatory variables. We assume that the signs for these variables are negative. The main stations are chosen as one station in each prefecture in terms of the number of passengers and the connection to Shinkansen stations. The chosen stations are Sapporo, Sendai, Mito, Utsunomiya, Omiya, Chiba, Tokyo, Yokohama, Nigata, Gifu, Nagoya, Tsu, Kyoto, Osaka, Sannomiya, and Hakata.

(9), (10) Regulation value of the floor area ratios and regulation value of the building coverage ratios are used as explanatory variables. When the regulation value of floor area ratio is high, land price can increase because landowners can build a high building. On the other hand, buildings surrounding the land can also be high because floor area ratio is common in a certain area. High buildings can block sunlight to the building, thereby decreasing the land prices. Regarding the regulation value of building coverage ratio, when this ratio is high, land price can increase because landowners can build large buildings. But the area with large buildings is unfavorable from the view-point of landscape. Accordingly, the signs for the floor area ratio and the building cov-erage ratio can be either positive or negative.

(11) Types of restrictions on land use are used as explanatory variables. To take account of heterogenous impacts of different restrictions on land use, we consider seven types of restrictions on land use which are imposed in residential areas. We show the seven land use districts in Table 3. 

(12), (13) Esthetic landscapes can increase land prices, so we use “landscape plan-ning area” and “landscape emphasis area” as dummy variables. A landscape planning area has restrictions on building heights and designs to protect esthetic landscapes. A landscape emphasis area, which is set within a landscape planning area, has a unique landscape, and is an area which should be well protected.

(14), (15) Newly developed residential areas can provide good environments. So, the land prices tend to be higher than in other areas. On the other hand, areas which were developed a long time ago normally have a high proportion of elderly residents and many vacant houses, which have negative effects. To account for these effects, we set dummies for newly developed areas and old developed areas. We define the areas developed before 1976 as the “older developed areas”, whereas the areas developed after 1976 as the “newly developed areas”. However, it is hard to know how large each newly developed residential area is. So, since the median value of Japanese developed residential areas is 67 ha, we assume that all the developed areas have a 460m radius. A circle with this radius has an area of about 67 ha.

(16) The road width dummy variable is used as an explanatory variable. It is defined whether road width is narrow (less than 5.5 meters), medium (between 5.5 and 13 ma-ter), or wide (more than 13 meters).

(17) Municipality dummy variables are used as explanatory variables in order to take account of municipality-specific factors which cannot be explained by other ex-planatory variables.

(18), (19), (20), (21), (22) Dummies representing whether the utility lines on the road are undergrounded or not, whether the road is in the neighborhood of the road with undergrounded lines, whether the road is narrow or wide, whether floor area ratio is more than 2 or not, and where the road is, are used as explanatory variables. Note that we already define the neighborhood in the third paragraph of Section 3.4. We have al-ready defined these variables in detail in Section 3.1.”

A.2 In Section 4.1., the authors choose the most reliable functional form depending on the P value in the hedonic estimations. However, I am concerned that the functional form, which was not selected due to the high P-value, could be a true model. In that case, some characteristics of the road with underground utility lines may not affect their price. It would be helpful to explain why this paper's method of choosing functional forms is appropriate.

[Response]

We understand your concerns. Actually, the number of samples (i.e., roads) without undergrounding is quite large compared to the number of samples with undergrounding. Accordingly, the R squared to three decimal places is constant regardless of the change in the threshold in FAR. In addition to the R squared, we have added AIC. Using the R squared and the AIC, we have explored which functional form and which threshold of FAR are best, and have added the discussion.

B1. When I substituted Equation (7) to Equation (2), I get the following result, which is slightly different from Equation (8). It would be helpful to explain how to derive the right-hand side of Equation (8).

[Response]

In the original, we used “applying eq.(7) to eq. (2)”. This was vague. In the revised version, we have used “Substituting the left-hand side of eq. (7) into the right-hand side of eq. (2)”. This directly yields eq. (8).

B.2 It would be helpful to include the neighbourhood dummy variable in Equations (9)-(12).

[Response]

As the reviewer suggests, we added a dummy variable in Eq. (12), which represents the roads in the neighborhood not facing roads with undergrounded lines.

B.3 In Table 4, do the authors use “FAR threshold” to set the building height dummy variable? I was confused with the use of this threshold. It would be helpful to clarify it.

[Response]

This should be explained. Actually, we have noted the reason why we use the FAR threshold to set the threshold. “We explore which FAR is a threshold differentiating the level of benefit of undergrounding utility lines. For this treatment, it can be argued that FAR regulation cannot completely represent the real building sizes. This is true. However, we use this FAR regulation for the following reasons. There are no other appropriate data for representing the real building sizes. Actually, in Japan, one source provides building sizes at each plot. However, the information is not complete. For example, the heights of some buildings of one or two stories are not given. In addition, some lots have no height data. Accordingly, we cannot completely match the heights of the buildings and the benefits of undergrounding utility lines.

Compared to the insufficient data of building heights, the data on FAR regulations is complete. As we show later, the estimates using the data on FAR regulations show clear results. These clear results show that real building sizes reflect the regulated FAR. Actually, it is often said that regulations on FAR follow the market equilibrium FAR in practical situations. In addition, we use only the average height of the buildings on a road (not the heights of a specific building) because our data unit is a road. So, this setting does not generate a large bias in estimation of the effect of the average height on the landscape.”

B.4 Footnote 13 is not easy to follow. Implementing Mapion URL is mandatory for who?

[Response]

“Implementing Mapion URL” is mandatory for mapping makers. But, in the first round, they require me to remove all footnotes. This footnote is not important for our manuscript. So, we removed this text.

B.5 In Lines 347-348, the authors use the term “the parameter for undergrounded lines with medium and wide roads and high buildings.” I suggest replacing it with the term “the parameter for undergrounded lines with medium or wide roads and high buildings.”

[Response]

We have followed the suggestion.

B.6. I understand that the WTP in Table 6 is estimated using the results in Table 4. The building height is classified as "Narrow" or "Wide" in Table 6 but is classified as "Narrow" or "Medium/Wide" in Table 4. It would be better to be consistent in its classification.

[Response]

We agree with this. We have followed this suggestion.

B.7. What is the data source of the floor area in Table 6?

[Response]

Actually, because this data is publicly released, there are several sources for this. But we use the data from the same source of land price data, which is National Land Numerical Information. We have put this in No. 21 of Table 3.

Reviewer 3 Report

This manuscript explores the benefits of burying overhead cables underground in the Japanese context. It derives a direct relationship between the move of burying utility lines underground and the change of land price. Also, the willingness to pay (WTP) can also be computed based on land price, which is different from the commonly used survey method via questionnaires. The analysis results are proved to be compatible with the related existing studies. Overall, this manuscript is well written. However, there are still some points that can be improved.

(1) Other evaluation studies related to the external benefits of underground utility tunnels or similar projects should also be reviewed.

(2) Line 47, there should be more detailed justifications on why we should consider the external benefits against high costs.

(3) The title of Section 2: is there anything to do with hedonic approach in this section?

(4) Line 97 and other places: should it be “goods” consumption?

(5) Line 105, why is this the optimal condition for households?

(6) From Eq.(7) to Eq.(8), how is it transformed? Shouldn’t the h be equalized by F/n?

(7) The symbol of “h” in Section 2 can easily be confused with the height symbol in Section. It would be better to change it into a more common denotation symbol for lot size.

(8) The title of Section 3 can be considered to change into “empirical model and data”.

(9) The K in Eqs (8)-(12) should be lower-case k.

(10) Line 22, (11) should be Eq. (11).

(11) Line 294-297, it is difficult to follow. It could be much easier and better to explain with an illustrating figure.

(12) Table 2, how to define low and high buildings at this stage?

(13) Table 3 No. (10), how are the data collected?

(14) Line 418, symbols are not explained. It is not clear how ΔP is derived.

(15) Line 439-440, Is it what you mean? Seems should be solid lines for cases with underground utility lines and dashed lines for cases without.

(16) Line 465, there are changes but not obvious?

(17) Line 470-471, what is the basis that the several million yen is calculated on?

(18) There should be a separate section to discuss the shortcomings of the method. Also, a comparison between the method in this study and the general survey method via questionnaire to get WTP should be included in the discussion.

(19) It would be helpful if some planning implications or detailed urban development guidelines regarding utilities lines can be put forward in the concluding remarks or elsewhere.

Author Response

Thank you for your detailed comments and suggestions. To respond to your comments, we revised our paper as follows. We will show our responses to your comments one by one.

(1) Other evaluation studies related to the external benefits of underground utility tunnels or similar projects should also be reviewed.

[Response]

As you pointed out, we have only six papers related to cost-benefit analyses of undergrounding utility lines. However, we have spent a lot of time searching for related papers, developing this paper for the last several years. For journal papers which can be captured by Scopus, we believe that we have reviewed most of the papers (some old papers are not cited on purpose). There might be other related documents in governmental reports, but it is hard for us to collect them.

(2) Line 47, there should be more detailed justifications on why we should consider the external benefits against high costs.

[Response]

We have added to the end of this paragraph, “If we find that certain characteristics can enhance the benefits of undergrounding utility lines, we can use the found characteristics to efficiently select sites for undergrounding”.  

(3) The title of Section 2: is there anything to do with hedonic approach in this section?

[Response]

Yes. Section 2 involves the hedonic approach. Hedonic approach measures the benefits using land prices. Section 2 shows how we can capture the benefits using land prices. Traditional hedonic approaches assume that households live in detached houses. But land can be occupied by condominiums. In particular, our setting involves the heights of condominiums. Surprisingly, in the context of the hedonic approach, condominiums have not been explicitly taken into account. So, we develop a theoretical basis for measuring the benefits of undergrounding utility lines with the hedonic approach in this section. To clarify this, we added “benefits of” to the original section title.

(4) Line 97 and other places: should it be “goods” consumption?

[Response]

We have changed all instances of “good” to “goods”.

(5) Line 105, why is this the optimal condition for households?

[Response]

We have added “, which indicates the equality of the marginal substitution and the price,” after the condition.

(6) From Eq.(7) to Eq.(8), how is it transformed? Shouldn’t the h be equalized by F/n?

[Response]

We added “, which is equal to ” after explanation of  under eq. (7). (We changed the notation h to q, as shown below.)

(7) The symbol of “h” in Section 2 can easily be confused with the height symbol in Section. It would be better to change it into a more common denotation symbol for lot size.

[Response]

We have changed h to q.

(8) The title of Section 3 can be considered to change into “empirical model and data”

[Response]

We have changed the title as the reviewer suggested.

(9) The K in Eqs (8)-(12) should be lower-case k.

[Response]

Do you mean the K above Σ? If so, this K is correct. As shown below eq. (12), we consider multiple attributes. K is the last attribute.

(10) Line 22, (11) should be Eq. (11).

[Response]

We have revised the text as the reviewer suggested.

(11) Line 294-297, it is difficult to follow. It could be much easier and better to explain with an illustrating figure.

[Response]

In the revised version, we have inserted a graph to explain this.

(12) Table 2, how to define low and high buildings at this stage?

[Response]

We have added explanations here.

(13)Table 3 No. (10), how are the data collected?

[Response]

We have added explanations here in the revised version.

(14) Line 418, symbols are not explained. It is not clear how ΔP is derived.

[Response]

We have added explanations here in the revised version.

(15) Line 439-440, Is it what you mean? Seems should be solid lines for cases with underground utility lines and dashed lines for cases without.

[Response]

We have added explanations in the revised version. Specifically, under Fig.2, we show that the dashed lines imply the range of sky view without utility lines; the solid lines represent the range of sky view with utility lines.

(16) Line 465, there are changes but not obvious?

[Response]

Adding “significantly” to the original text, we have “There is very little difference between the WTP with high buildings and that with low buildings.”

(17)Line 470-471, what is the basis that the several million yen is calculated on?

[Response]

We have added “shown in Table 6” here.

(18) There should be a separate section to discuss the shortcomings of the method. Also, a comparison between the method in this study and the general survey method via questionnaire to get WTP should be included in the discussion.

[Response]

We have extended section 4.3, which was one paragraph in the previous version, to a new section 4.3 “Possible biases of estimation”. In the first paragraph in the new section, we discuss the comparison between the revealed preference methods and the stated preference methods. After this discussion, we discuss possible biases of the hedonic approach, which is a typical revealed preference method, in our context.

(19) It would be helpful if some planning implications or detailed urban development guidelines regarding utilities lines can be put forward in the concluding remarks or elsewhere.

[Response]

We think that our paper can be helpful in developing a method to measure the benefits of undergrounding utility lines. But proposing the method is not our purpose. When measuring the benefits in the real situation, they might have more information on scenery, residents’ incomes, or other relevant matters. We do not like to propose some inflexible methods. So we just put in the end of the first paragraph of the conclusion, “Specifically, when constructing a cost-benefit manual for undergrounding utility lines, the road width and the building heights should be taken into account.”

Round 2

Reviewer 1 Report

The following statement needs appropriate citation.

"However, as long as we measure the benefit using the hedonic approach, this reverse causality has no problem. Regardless of the reasons for undergrounding projects, when lines are undergrounded at a certain location, the land prices around the location increase. The increases in land price can be captured by land price functions as long as they appropriately control for relevant explanatory variables. This point is an advantage of the hedonic approach."

Author Response

The statement as a whole is original. But an increase in land price due to amenity improvements is called “capitalization”. In the revised version, we have referred to this “capitalization” and cited Kanemoto (1988). Kanemoto (1988) proves that perfect mobility between different areas ensures that property prices reflect the benefits of improved amenities. This is called capitalization. So, we have changed our original text to “However, as long as we measure the benefit using the hedonic approach, this reverse causality has no problem. Regardless of the reasons for undergrounding projects, when lines are undergrounded at a certain location, the land prices around the location increase. Indeed, Kanemoto (1988) proves that perfect mobility between different areas ensures that property prices reflect the benefits of improved amenities, and calls this property ‘capitalization’. The increases in land price can be captured by land price functions as long as they appropriately control for relevant explanatory variables. This point is an advantage of the hedonic approach.”

Reviewer 3 Report

The authors did a very good revision to improve the quality of the manuscript. Even so, there are still some points to be further addressed. It can be accepted after the following checks and further considerations.

(1) Regarding the Response No.(2), these works can be used as evidence to justify the needs of considering external effects of underground infrastructures in the introduction:

Kaliampakos, D., Benardos, A., Mavrikos, A., 2016. A review on the economics of underground space utilization. Tunnelling and Underground Space Technology, 55, 236–244.

Mavrikos, A., Kaliampakos, D., 2021. An integrated methodology for estimating the value of underground space. Tunnelling and Underground Space Technology, 109, 103770.

Peng, F.L., Qiao, Y.K., Sabri, S., Atazadeh, B., Rajabifard, A., 2021. A Collaborative Approach for Urban Underground Space Development towards Sustainable Development Goals: Critical Dimensions and Future Directions. Frontiers of Structural and Civil Engineering, 15(1), 20–45.

Qiao, Y.K., Peng, F.L., Wang, Y., 2017. Monetary valuation of urban underground space: A critical issue for the decision-making of urban underground space development. Land Use Policy, 69, 12–24.

(2) Regarding the Response No.(7), some equations still has h, such as the ones in Line 116, 118, 128, etc. This is a critical issue that should be taken seriously.

Author Response

Reviewer's comments are denoted below and in blue, and our responses are put after [Response].

The authors did a very good revision to improve the quality of the manuscript. Even so, there are still some points to be further addressed. It can be accepted after the following checks and further considerations.

Regarding the Response No.(2), these works can be used as evidence to justify the needs of considering external effects of underground infrastructures in the introduction.

[Response]

Thank you for showing the previous research about the benefits of undergrounded infrastructure. We have mentioned all the papers in our Introduction.

Regarding the Response No.(7), some equations still has h, such as the ones in Line 116, 118, 128, etc.

[Response]

Thank you very much. In the previous version, we missed some of the instances of h when we replacing them with q.